# Silk Fibroin Conjugated with Heparin Promotes Epithelialization and Wound Healing

**DOI:** 10.3390/polym14173582

**Published:** 2022-08-30

**Authors:** Rikako Hama, Derya Aytemiz, Kelvin O. Moseti, Tsunenori Kameda, Yasumoto Nakazawa

**Affiliations:** 1Department of Biotechnology and Life Science, Graduate School of Engineering, Tokyo University of Agriculture and Technology, 2-24-16 Naka-Cho, Koganei 184-8588, Japan; 2Silk Materials Research Unit, Institute of Agrobiological Sciences, National Agriculture and Food Research Organization, 1-2 Owashi, Tsukuba 305-8634, Japan; 3National Sericulture Research Centre, Industrial Crops Research Institute, Kenya Agricultural and Livestock Research Organization, Thika P.O. Box 7816-01000, Kenya

**Keywords:** silk fibroin, heparin, tissue engineering, wound healing, dressing material

## Abstract

Silk fibroin (SF) has attracted attention as a base biomaterial that could be suitable in many applications because of its shape and structure. Highly functional SF has been developed to promote tissue regeneration with heparin conjugation. However, the hydrophobic three-dimensional structure of SF makes it difficult to bind to high-molecular-weight and hydrophilic compounds such as heparin. In this study, sufficient heparin modification was achieved using tyrosine residues as reaction points to improve cellular response. As it was considered that there was a trade-off between the improvement of water wettability and cell responsiveness induced by heparin modification, influences on the structure, and mechanical properties, the structure and physical properties of the SF conjugated with heparin were extensively evaluated. Results showed that increased amounts of heparin modification raised heparin content and water wettability on film surfaces even though SF formation was not inhibited. In addition, the proliferation of endothelial cells and fibroblasts were enhanced when a surface with sufficient heparin assumed its potential in assisting wound healing. This research emphasizes the importance of material design focusing on the crystal structure inherent in SF in the development of functionalized SF materials.

## 1. Introduction

Skin damage is often caused by physicochemical factors, such as trauma, surgery, and diabetes. Typically, skin tissue has a remarkable ability to restore and regenerate. However, if the level of damage is significant or regeneration is lowered, inadequate tissue restoration can reduce the patient’s quality of life. Wound dressings provide a moist environment suitable for wound care and protection from external stimuli [1,2]. In recent years, various studies have been conducted in tissue engineering, focusing on dressings that decompose and get replaced with new tissue as tissue regeneration progresses. In other words, to achieve biological-level tissue regeneration, there is a need for wound dressings that minimize inflammatory response and promote rapid tissue formation and maturation.

Tissue engineering combines cells, scaffolds, and growth factors (GFs) to promote tissue regeneration [3,4]. Research on culturing epidermal cells collected from patients, processing them as sheets, and returning them to patients as implants has been put into practical use [5]. In addition, a more versatile method has been reported in research whereby placing and cultivating cells into a biomaterial which is then transplanted into an affected area [6,7]. However, these methods are unsuitable for treating acute injuries since seeding and culturing cells require time. One of the promising approaches is to apply GFs to a scaffold to increase the activity of recruited cells after transplantation into the wound. It has already been reported that the healing period can be further shortened by some GFs and their functional peptide, which helps promote epithelialization and granulation. However, the induction of tissue renewal at the native level is still challenging, and general-purpose materials for elderly patients, whose demand is increasing, have not been obtained.

We considered a noninvasive and general-purpose material concept that induces tissue regeneration by specifically capturing GFs secreted during the healing process and supplying them to migrated cells. Heparin, a component of the extracellular matrix, was chosen since it forms complexes by electrostatic interaction and acts on the binding of cells to GFs receptors [8] which regulate various cell functions such as cell growth, differentiation, morphology, and migration. It binds to various major GFs: vascular endothelial growth factor (VEGF) [9], fibroblast growth factor (FGF) [10,11], epidermal growth factor (EGF) [12], and hepatocyte growth factor (HGF) [13]. Thus, heparin was expected to act as a functional molecule that specifically captures secreted GFs into the material during the healing process. In addition, silk fibroin (SF), which has mild and controllable, biodegradabilty [14], exhibits low inflammation [15], and promotes skin regeneration [16,17,18], was selected as a base material. These are attractive properties for a tissue engineering material. Its application to the renewal of various tissues is being studied because it can be processed into various forms such as films, gels, and nonwoven fabrics [19,20,21]. These properties are based on SF being a hydrophobic crystalline protein. The highly aggregated β-sheet crystal structures formed mainly by the hydrophobic sequence of (Gly-Ala-Gly-Ala-Gly-Ser)_n_ are scattered in amorphous portions constituted of sequences with polar and bulky side chains such as tyrosine (Tyr) residues [22]. From this, it is possible to control the properties of SF from the induction of recrystallization, and it is more suitable than other biological polymers. Several studies have been published on combining heparin and SF [23,24,25]. Cell proliferation has been shown to improve on nonwoven fabrics by the electrospinning of heparin and SF solutions [26], and a sponge with heparin physically immobilized on the SF has been reported [25]. In addition, the application of a hydrogel to a skin defect promoted wound closure [23]. However, the material’s mechanical properties should be comparable to normal tissues because they are affected by blood and mechanical stress due to daily life during the healing period. In many cases, it is difficult to assess from its properties whether a sufficient amount of heparin is modified because SF after heparin modification is affected by the high hydrophilicity of heparin, and its stability to water decreases.

Therefore, it was hypothesized that the influence on the characteristics of SF based on the crystal structure formation could be reduced by selecting the Tyr residue localized in the semi-crystal region as the reaction point. The highest amount of reactive amino acids available for chemical modification are Serine (Ser) residues at 12.1%, and Tyr residues at 5.3% [27]. Unlike the Ser residues, the Tyr residues are localized outside the crystal region and as such, it is possible to secure sufficient reaction points for modification without a significant loss of mechanical properties caused by the inhibition of β-sheet crystal structure formation. However, the effect of steric hindrance on modifying polysaccharides with high hydrophilicity and a very high molecular weight on secondary structure formation cannot be ruled out. To consider this hypothesis, we prepared materials with varying amounts of heparin modification and evaluated the effect of this modification on the structure of SF and its stability in water by comparing these materials. In addition, the interaction with epithelial cells and fibroblasts was evaluated for application to skin wound dressings.

## 2. Materials and Methods

### 2.1. Preparation of SF Aqueous Solution

As previously described, raw silk fibers were obtained from *Bombyx mori* cocoons by reeling and drying [28]. Then, 250 g of dry raw silk fibers were placed in 0.02 M sodium carbonate aqueous solution at 5 °C for 30 min to degum sericin. The degummed SF were then rinsed with purified water several times and dried at room temperature. Next, the SF fibers were dissolved in 9 M aqueous lithium bromide (LiBr, FUJIFILM Wako Pure Chemical Cor., Osaka, Japan) at 37 °C for 2 h to obtain an aqueous LiBr solution of SF. This solution was dialyzed against distilled water until all the salts were removed and then centrifuged. The final concentration of the purified SF aqueous solution was recorded as approximately 4.5 wt% by measuring the weight of the substrate obtained by drying the SF solution after the end of dialysis.

### 2.2. Modification of SF with Heparin

Heparin binding to SF was performed using different amounts of heparin [29,30]. Reaction points in SF and the molar ratio of each reagent used to prepare each sample were as shown in Table 1. Figure 1 shows the scheme of the modification reaction which was calculated from the amount of Tyr and Lys residues in the SF heavy chain and the average molecular weight of heparin (15,000 Da). Briefly, aqueous heparin solutions were prepared by adding 576, 864, 1152, and 1440 mg of heparin sodium (FUJIFILM Wako Pure Chemical Co., Osaka, Japan) to sodium carbonate, in 18 mL of purified water to attain a pH over 10. A solution of 2,4,6-trichloro-1,3,5-triazine (Cyanuric Chloride, CY, Kanto Chemical CO., INC., Tokyo, Japan) was prepared by dissolving 72 mg in 6.0 mL of 1,4-dioxane (Sigma-Aldrich, Saint Louis, MO, USA). While stirring the heparin aqueous solution over an ice bath, CY solution was added dropwise at 24 mL/h using a syringe pump. The heparin–CY solution reaction proceeded for 2 h under ice-cold conditions. After 2 h, 24 mL of a 2.0 wt% SF solution was added to the heparin–CY conjugated solution. The reaction was performed at 37 °C for 24 h and was stopped by adding 1 M HCl, transferred to a cellulose tube (size 21.4, Nihon Medical Science, Osaka, Japan), and dialyzed against purified water. The dialysis was continued for more than five days to remove unreacted substances and solvents. The dialyzed reaction solution was then centrifuged at 8500 rpm for 30 min to remove the residue and obtain an aqueous solution of HSF.

### 2.3. SF-Heparin Reaction Confirmation by Solution-State ^1^H NMR

The progress of the SF heparin conjugation reaction was confirmed by NMR spectroscopy [31]. Each SF and HSF solution was diluted to concentrations of 0.5 wt% with D_2_O. ^1^H NMR spectra were acquired on a JEOL ECA 500 spectrometer (JEOL Resonance, Tokyo, Japan) operating at 500 MHz for ^1^H observation using a 5 mm probe. Water pre-saturation (watergate pulse [wgh]) sequence and DPFGSE spectra were applied. Spectra were recorded with a 5000 Hz spectral width and a 5.0 s relaxation delay. Fourier transform was performed using the processing program to obtain ^1^H NMR spectra from free induction decay.

### 2.4. Secondary Structure Analysis by Solid-State ^13^C CP/MAS NMR

Solid-state NMR measurements of films, before (as-cast) and after insolubilization with high humidity, were performed to examine the changes in the secondary structure of SF in HSF [32]. In these experiments, SF and HSF5 with the highest heparin modification were selected. Thick cast films for film characterization were obtained by the following procedure: lyophilized sponges of SF and HSF aqueous solutions were dissolved in 1,1,1,3,3,3-hexafluoro-2-propanol (HFIP) at 6.0 wt%, and 3.0 mL cast on Teflon dishes with a diameter of 50 mm and air-dried, then incubated in a container with about 100% relative humidity at 37 °C for 24 h. The solid state ^13^C CP/MAS NMR measurement used the Solid-state NMR equipment by AVANCE WB 400 (Bruker Japan K.K., Yokohama, Japan) at a spinning speed of 9 kHz and 12,288 accumulations with a resonance frequency of 400 MHz. The contact time was 5.00 ms, ^1^H 90° pulse (3.20 μs), and a pulse delay of 6.0 s. 

### 2.5. Evaluation of Film Surface Properties

HSF aqueous solutions were cast on glass plates and air dried at room temperature, then incubated in high humidity at 37 °C for insolubilization for 24 h. Elements on insolubilized film surfaces were evaluated by SEM–EDS analysis with an energy dispersive X-ray analyzer Quantax 70, using a scanning electron microscopy (SEM) Miniscope^® TM^ 3000 (Hitachi High-Technologies, Corp., Tokyo, Japan). Elemental analysis was performed for sulfur (S) and sodium (Na) atoms derived from heparin, and their distribution on the film surface was determined.

For water contact angle measurements, thick cast films were made following a procedure to set the instrument. Each HSF sponge was dissolved in 98% formic acid (FA, FUJIFILM Wako Pure Chemical Co., Osaka, Japan) at 6.0 wt%, then 3.0 mL of each solution was cast on polystyrene dishes and air-dried at room temperature. The cast films were incubated at 37 °C and 100% relative humidity for 24 h for insolubilization, hollowed out, and pasted on a glass substrate with a double-sided tape. The contact angle of the film surface was calculated according to the θ/2 method from the photograph of the droplets taken 1 s after dropping 2 μL of ultrapure water with a DMo-501Hi dispenser (Kyowa Interface Science Co., Ltd., Niiza, Japan). Results were given as means ± standard derivation (*n* = 5).

### 2.6. Water Absorption

The water content of films (SF and HSFs) was measured. Each SF and HSF aqueous solution was cast on glass plates and air dried at room temperature. After high humidity insolubilization, films were vacuum dried for 3 h and measured as dry weight. Then they were immersed in purified water for 3 h and measured again as wet weight. The water content was calculated using Equation (1) below. The results were presented as means ± standard derivation (*n* = 3).
The percentage water content = [(wet weight − dry weight)/dry weight] × 100(1)

### 2.7. Dynamic Viscoelasticity Evaluation

To evaluate the effects of heparin modification on the mechanical properties of SF, dynamic viscosity analysis (DVA) in air and water was performed using a DVA-205 (IT Metrology Control Co., Ltd., Osaka, Japan) instrument. For this evaluation, thick cast films (SF, HSF/FA) were used, as described above. Measurements in the air were performed in the tensile mode at a constant temperature rise to calculate the average storage modulus, modulus loss, and loss coefficient at 37 ± 0.2 °C. Each sample was cut into a rectangle sample of 40 mm × 5 mm and evaluated at a frequency of 1 Hz, with a 30 mm distance between the gripper tools. A film immersed in purified water for 3 h was used for underwater measurements. According to the manufacturer’s protocol, the evaluation was performed using a rectangular sample (25 mm × 5 mm). The mean storage modulus, modulus loss, and modulus of loss were also calculated from the mean values measured for 10 min in the tensile mode. The measurement conditions were a water temperature of 37 ± 0.2 °C, a frequency of 1 Hz, and a distance between the gripper tools of 15 mm. Results were given as means ± standard derivation (*n* = 3).

### 2.8. Cell Culture

HUVEC and NHDF lines (Lonza, Walkersville, MD, USA) were purchased. HUVECs were grown in HUVEC-XL™ cell systems EBM™ 2 (Lonza) that contained 2% (*v/v*) fetal bovine serum (FBS), 0.5 mL vascular endothelial growth factor (VEGF), 0.5 mL human epidermal growth factor (hEGF), 0.5 mL insulin-like growth factor-1 with the substitution of arginine (Arg) for glutamic acid (Glu) at position-3 (R3-IGF-1), 0.5 mL ascorbic acid, 0.2 mL hydrocortisone, 2 mL human fibroblast growth factor-base (hFGF-β), 0.5 mL gentamicin/amphotericin-B (GA), and 0.5 mL heparin. NHDFs were cultured with an FBM-BulletKit™ (Lonza) supplemented with 2% (*v/v*) FBS, 0.5 mL hFGF-β, 0.5 mL insulin, and 0.5 mL GA. The cells were maintained in a humidified 5% CO_2_ atmosphere at 37 °C. In the HUVEC experiment, heparin was excluded from the kit to avoid a cross reaction. The passages used for experiments of HUVEC were three and four, while that of NHDF was six.

### 2.9. Evaluation of Cell Proliferation

Cellular responses to each HSF sample were evaluated on cast films. Each aqueous cast film of SF and HSFs was prepared in 24-well plates at 0.25 mg/cm^2^, immersed in 70% (*v/v*) ethanol solution for 30 min for insolubilization and sterilization, and then thoroughly washed with ultrapure water. HUVEC was seeded on films swollen with PBS at a density of 1.0 × 10^3^ cells/mL, whereas NHDF was seeded at 0.5 × 10^3^ cells/mL. The cell cultures were maintained at 37 °C in a 5.0% CO_2_ atmosphere, and each medium was replaced with a fresh one every two or three days. NHDF was incubated for up to seven days to investigate long-term growth behaviors, while HUVEC was incubated for three days. The MTS assay was used to evaluate cell proliferation by measuring the metabolic activity of the cells. Specifically, 20 μL of MTS reagent prepared with a volume ratio of MTS:PMS = 20:1 was added to 100 μL of fresh medium. The metabolic activity of the cells produced the dye during the 2 h culture. The mixture was transferred to a 96-well plate, and absorbance at 492 nm was measured. Uncoated wells (TCP) were used as positive controls, while unseeded wells with medium and reagents were used for blank measurements. Results were given as means ± standard derivation (*n* = 6).

### 2.10. Statistical Analysis

All data are presented as mean ± standard deviation. One-way ANOVA and Tukey’s statistical analyses were performed by multiple comparisons using Origin software (OriginLab, Northampton, MA, USA). A probability value (*p*) of less than 0.05 was considered statistically significant (* *p* < 0.05).

## 3. Results

### 3.1. SF-Heparin Conjugation Reaction Confirmation

After dialysis, all HSF solutions were colorless and transparent. The concentrations of HSFs were between 0.5 and 1.0 wt%. Figure 2a shows the chemical shift in the ^1^H NMR spectrum of pure SF, heparin, and each HSF sample obtained to measure the progress of the heparin-Tyr residue reaction. For example, a characteristic peak derived from the alanine (Ala) residue of SF (H_β_), which was nearly 1.2 ppm, and a characteristic peak derived from the heparin acetyl group (CH_3_), nearly 1.9 ppm [33], were confirmed in each spectrum of the HSF samples. The superimposed spectra of the SF Tyr residues, H_δ_ and H_ε_, were as shown in Figure 2b. Peaks derived from aromatic rings H_δ_ and H_ε_ in Tyr residues appeared in pure SF and each HSF sample. In the pure SF spectrum, the peak at 6.65 ppm was derived from H_ε_, and the peak at 6.93 ppm was derived from H_δ_. From the expanded spectrum of the Tyr residue, we found that the peak derived from H_ε_ disappeared after the conjugation reaction, and a new peak appeared at 7.10 ppm. This means that the CY used in the modification reaction reacted with the hydroxy group of the Tyr residue [29]. The chemical shift value of the peak changed due to a change in the local electron density.

In addition, Ser residue H_β_ and glycine (Gly) residue H_α_ peaks from SF with the hydroxyl group peak contained in heparin appeared in the expanded spectrum (from 2.5 to 4.5 ppm: Figure 2c). Notably, the HSF spectra’s SF-derived peaks were superimposed at different intensities. Further, we observed that the intensity of the heparin-derived peak (Heparin CH, around 3.5 ppm) changed based on the amount of heparin used in the modification reaction.

### 3.2. Secondary Structure Analysis

The solid-state ^13^C CP/MAS NMR spectra of the HSF film before and after the insolubilization treatment are shown in Figure 3. Among these spectra, most attention is paid to the peak derived from the Ala residue C_β_ appearing between 10 and 30 ppm, making it easy to evaluate the secondary structure of SF [32,34]. In the insolubilized SF (SF-T) spectrum, the peak at 19.2 ppm indicates that the β-sheet structure appeared more predominantly than the peak at 14.8 ppm, indicating the random coil. Focusing on the spectrum of HSF5-N before insolubilization, the peak derived from the acetyl methyl group of heparins at 22.5 ppm and the peak of the random coil at 14.8 ppm were confirmed. After insolubilization, the HSF5-T spectrum indicated the presence of β-sheet structure, as in SF-T. Thus, it was confirmed that the structural change of SF by the insolubilization treatment was successful.

### 3.3. Heparin Coverage of Film Surface

For elemental analysis of the film surfaces by EDS, SEM images of both the HSF and SF films were taken (Figure 4). It was confirmed that the distribution of S and Na, derived from the sodium sulfate of heparin, on the film surface became denser as the amount of heparin used in the modification reaction increased. However, despite the fact that the highest amount of heparin was used in the reaction to prepare the HSF5 sample, there was no high qualitative difference between the mass ratios of S and Na in HSF5 and HSF4. Further, although the presence of heparin on the film surface was confirmed, it could not be evaluated quantitatively because of the low resolution of the measuring instrument.

The hydrophilicity of the film surfaces was also evaluated based on water contact angle measurements. As the amount of heparin modification increased, the hydrophilicity of the SF significantly improved (Figure 4f). On the other hand, no significant increase in hydrophilicity was confirmed in HSF2, which contained the lowest amount of heparin. However, as the amount of modification increased, the contact angle improved and approached between 60° and 70°, which was suitable for the initial adhesion of cells.

### 3.4. Evaluation of Water Absorption

Figure 5 shows the moisture content calculated from each HSF and SF film’s wet-to-dry weight ratio. HSF4 and HSF5 films had significantly higher water content than SF films: a maximum water content approximately 15 times that of SF was obtained. On the other hand, there was no significant difference between the water contents of HSF2 and the SF films. This tendency was similar to that of the contact angle results, indicating that heparin modification at least more than HSF2 is necessary to improve the water properties of SF. By improving the water content, which is deficient in SF, it is expected that it will absorb more GF-rich exudate when used in a biological healing environment, that is, it will be helpful in efficiently capturing GFs.

### 3.5. Dynamic Viscoelasticity Evaluation

The storage elastic modulus (E′) and loss elastic modulus (E″) of each of the HSF and SF films obtained from dynamic viscoelasticity measurements in air or water are shown in Figure 6a,c, and the loss coefficient (tan δ) is shown in Figure 6b,d. In the dry state, the observed storage and loss modulus results were comparable to SF in any HSF, indicating that the mechanical properties of SF were preserved.

On the other hand, the results in the wet state showed significantly lower values for all HSF films than for SF films, improving the flexibility of the films. Looking at the loss factor, which indicates the material’s viscous properties, the viscoelasticity of SF and HSF tends to be reversed in the dry and water content states.

### 3.6. Cell Proliferation In Vitro

Figure 7 shows the results of the MTS cell proliferation assay. Cell proliferation was higher in all HSF samples 24 h after culture than in the positive controls, TCP and SF. In addition, this tendency continued even on the third day of culture, and the values of HSF3, HSF4, and HSF5, which had excellent water characteristics, were significantly higher. Further, fibroblasts, which play a significant role in forming granulation tissue neoplasms, were used to evaluate the long-term cell proliferation behavior. Fibroblast proliferation was also significantly enhanced at HSF3, HSF4, and HSF5 compared with TCP and SF. The timing at which proliferation is significantly promoted differs depending on the cell type. Altogether, the results indicate that HSF induced long-term improvement in cell activity for multiple cell types.

## 4. Discussion

This study developed a heparin-conjugated SF-based (HSF) material that promotes wound healing. HSF induces vascular endothelial cells and fibroblasts proliferation, a process necessary in tissue regeneration. We assessed the effects of HSFs on SF secondary structure transition, physical properties, and cell proliferation.

SF is a hydrophobic biopolymer consisting of a crystal region composed of a repeating sequence of (GAGAGS)_n_ and a quasicrystal region composed of a sequence of (GX)_m_GY)_n_ (X = A or V) [22]. When SF is placed in an organic solvent or under high humidity conditions, a tightly packed β-sheet crystal structure is induced, which shows toughness and water insolubility characteristics. Previous researchers have integrated polysaccharides into SF scaffolds [23]; however, polysaccharides such as heparin, with considerable molecular weight and high hydrophilicity, disturb this characteristic crystal structure formation. Furthermore, the resulting conformation sterically hinders heparin-SF conjugation and could be an obstacle. Therefore, Tyr residues, an abundant amino acid in the quasicrystal region, were conjugated with heparin, as they contribute minimally to the formation of the crystal structure. Still, we speculated that excessive heparin modification would inhibit the crystal structure transition. Therefore, we hypothesized an equilibrium point in modifying the underlying SF’s physical characteristics and prepared HSF samples containing varying amounts of heparin.

By ^1^H NMR measurement in the solution state, SF and heparin coexisted in each HSF sample. Theoretically, two SF side-chain amino acids reacted with cyanuric chloride: the phenolic hydroxyl group of the Tyr residue and the e-amino group of Lys. However, considering the effects of silk fiber refining, the abundance of Lys residues is much lower than that of Tyr residues in SF and is mainly bound via Tyr residues. In addition, we found that the intensity of the peak of heparin CH increased as the amount of heparin added to the conjugation reaction increased. Therefore, we demonstrated that the amount of heparin immobilized on SF could be adjusted by simply changing the amount of reagent used. Nonwoven or spongy fabrics are desirable skin wound dressings because they mimic the extracellular matrix. We selected films to evaluate the potential wound healing characteristics of high-performance SF. Random coils are predominant in the secondary structure of SF aqueous solutions obtained by dissolving SF fibers. The SF films were treated with high humidity to induce the β-sheet crystal structure conformation, making the SF films water-insoluble. ^13^C CP/MAS NMR measurements of HSF5, which had the highest amount of heparin of all the films, confirmed that the dominant secondary structure of SF changed after insolubilization. Thus, SF crystal structure formation was inhibited even when a high-molecular-weight hydrophilic polysaccharide was conjugated to hydrophobic SF. The morphology was sufficiently maintained in all the HSF films used in the wet state of the physical characteristics test.

The elemental analysis results by SEM-EDS and water contact angle measurements are well linked, reflecting the difference in the heparin modification rate even after processing into the film. Further, it is suggested that heparin and SF are unlikely to be separated and unevenly distributed even though there is a significant difference in hydrophilicity. Typically, SF is hydrophobic due to its very dense crystal structure. However, it is considered that the presence of heparin between the crystal structures improved the hydrophilicity on the film surface and inside the material. Furthermore, it is beneficial for improving hydrophilicity and cell responsiveness by capturing GFs secreted during the healing process. Dynamic viscoelasticity measurements showed that in the dry state, the mechanical properties of HSF were similar to SF, regardless of the amount of heparin modification. Focusing on the loss factor, SF exhibited the most robust elastic properties based on the uniformity of the crystal structure. Interestingly, however, HSF4 and HSF5, which have high water content, exhibited more robust elastic properties than SF underwater conditions. This could be due to the effect of water molecules in the HSFs interacting with the material and may be revealed by a detailed analysis such as thermal analysis and solid-state NMR measurements [35].

HSF significantly promoted the proliferation of endothelial cells and fibroblasts compared to SF. Good wound dressings should provide cells with mechanical support and induce tissue renewal. The surface hydrophilization of SF had a water contact angle close to 60–70°, a range suitable for cell adhesion via the adsorption of blood proteins. Furthermore, it is thought that cell proliferation was strongly promoted by the active capture and donation of GFs, such as FGF and VEGF. The above results clarified that heparin-modified SF promoted the proliferation of cells that regenerate tissues regardless of the type. In addition, it is believed that SF needs to be modified by heparin at least as much as the HSF3 sample for it to have such cellular responsiveness.

## 5. Conclusions

HSF in the dry state shows similar three-dimensional structure changes to SF and shows the same elastic modulus regardless of the amount of heparin modification. However, the water content was strongly dependent on the abundance of heparin, as was the hydrophilicity of the film surface. These increases in hydrophilicity on the surface and inside the material were expected to work not only to improve the adhesion of migrating cells to the material but also to increase the adsorption capacity of GFs through exudate absorption and promote the proliferation of vascular endothelial cells and fibroblasts. In this study, we conducted a wide-range evaluation focusing on the relationship between the material and water for HSF films that are stable in water and whose modification amount is gradually changed. It is possible to develop functional materials whose physical properties and cellular responsiveness can be more precisely controlled by focusing on modifying the crystal structure characteristic of SF. This provides the basic knowledge for enabling design toward induction material-driven wound healing.

## Figures and Tables

**Figure 1 polymers-14-03582-f001:**
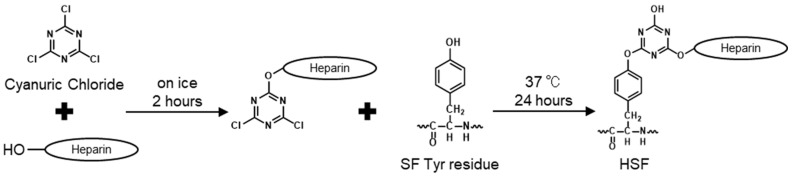
A schematic diagram of the chemical modification reaction of SF with cyanuric chloride–activated heparin.

**Figure 2 polymers-14-03582-f002:**
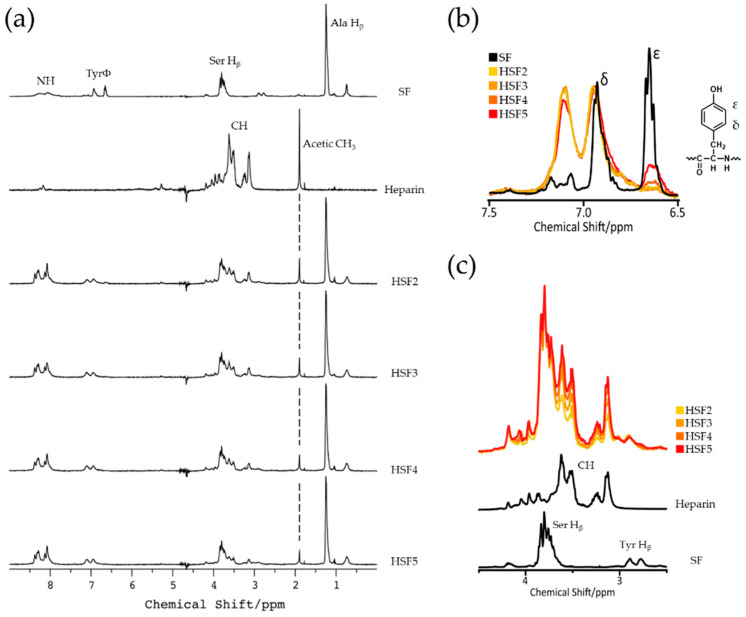
^1^H NMR spectra of SF and HSF aqueous solutions. Every concentration was 0.5 wt% and peak intensities were normalized by SF Ala H_β_. (**a**) SF, heparin, and heparin-modified SF samples (HSF2, HSF3, HSF4, and HSF5) from top to bottom. (**b**) Spectra around SF Tyr residues with a structural formula of a Tyr residue. (**c**) Spectra around a CH group region which contains heparin CH, SF Se) and Tyr residues demonstrated with different colors.

**Figure 3 polymers-14-03582-f003:**
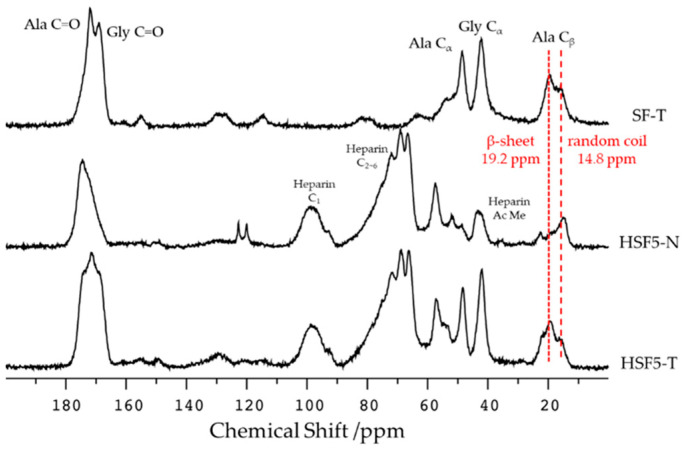
^13^C CP/MAS NMR spectra of film samples of SF-T, HSF5-N, and HSF5-T films (T: after insolubilized, and N: before insolubilized). Peak intensities were normalized by SF glycine C_α_.

**Figure 4 polymers-14-03582-f004:**
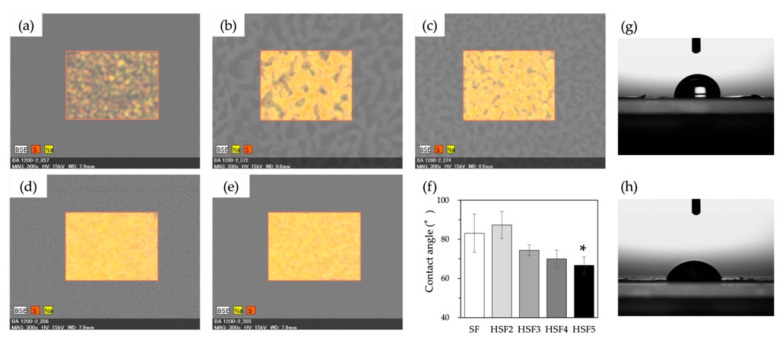
SEM-EDS images of the distributions of S and Na on the surface of SF (**a**), HSF2 (**b**), HSF3 (**c**), HSF4 (**d**), and HSF5 (**e**) films. (**f**) shows a water contact angle of SF and HSF samples ((**g**): SF, (**h**): HSF5). * *p* < 0.05 relative to SF.

**Figure 5 polymers-14-03582-f005:**
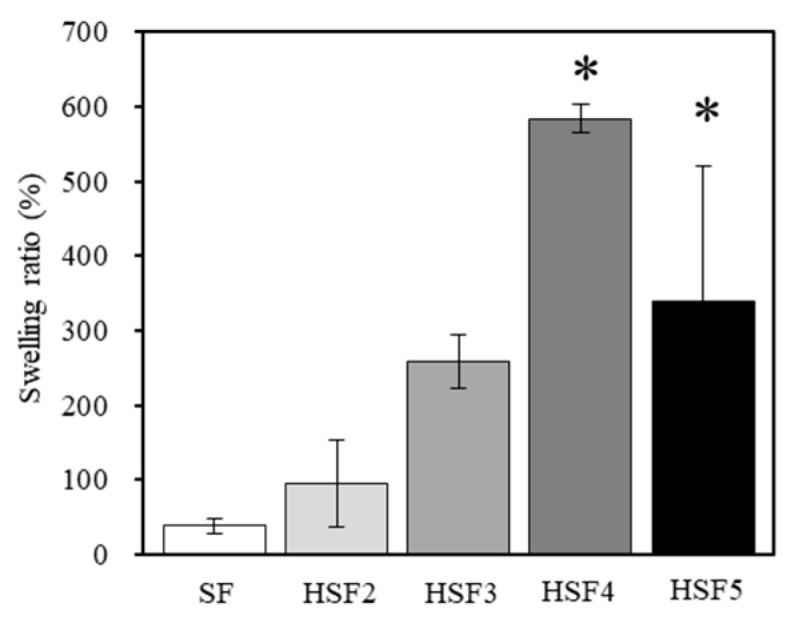
The swelling ratios of SF and heparin modified (HSF2, HSF3, HSF4, and HSF5) samples (* *p* < 0.05 relative to SF).

**Figure 6 polymers-14-03582-f006:**
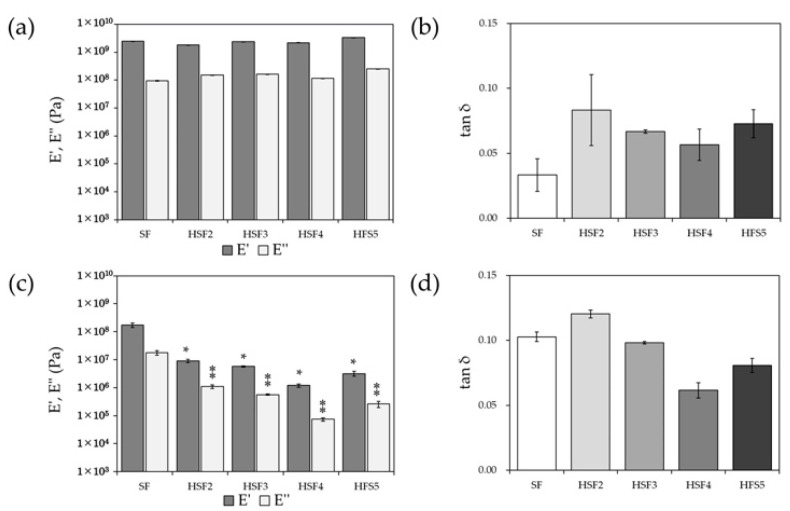
The storage elastic modulus (E′) and loss elastic modulus (E″) (**a**,**c**), and the loss factor (tan δ) (**b**,**d**) of SF and heparin-modified samples. (**a**,**b**) dry state and (**c**,**d**) wet state (*, ⁑ *p* < 0.05 relative to each SF).

**Figure 7 polymers-14-03582-f007:**
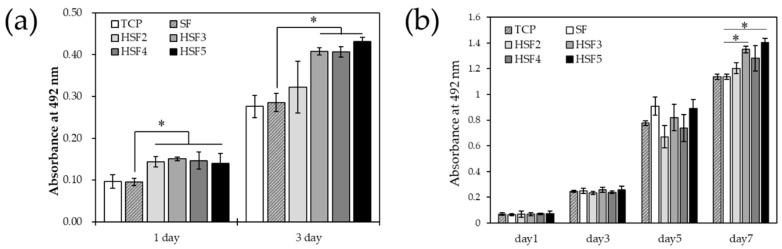
The proliferation of (**a**) HUVEC and (**b**) NHDF on a tissue culture plate (TCP), SF, and heparin-modified samples (HSF2, HSF3, HSF4 and HSF5) by MTS assay (* *p* < 0.05).

**Table 1 polymers-14-03582-t001:** Reaction points in SF and the molar ratio of each reagent used for the reaction.

	Target Sites in SF	Cyanuric Chloride	Heparin Sodium
HSF2	20	20	2
HSF3	20	20	3
HSF4	20	20	4
HSF5	20	20	5

## Data Availability

Not applicable.

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
