# Peer review of "Silk Fibroin Conjugated with Heparin Promotes Epithelialization and Wound Healing"

_polymers, 2022, doi:10.3390/polym14173582_

Round 1

Reviewer 1 Report

This is a good work, but needs to address the below issues to consider further.

Specific comments:

heparin exposed on the film surface correlated with the modification reaction, and water wettability: Provide the data here

. In addition, cell proliferation is promoted; Not clear which cells

 cell proliferation is promoted to the same extent: same extent compared to ?

with a sufficient amount of heparin, demonstrating its usefulness in assisting wound healing: Simply proliferation data did not claim the wound healing behaviour, should be evidenced by some other experiments like scratch assay and other experiments...

In applying SF in biomaterials: What do you mean biomaterials here? SF is a biomaterial

characteristic crystal structures from a multifaceted evaluation: Present the results in abstract.

research has been reported by: Give reference

was selected as the base material: In this study or earlier reference?

Several studies have been published on combining heparin and SF[21]: Provide a few more references here.

concentration of the purified SF aqueous solution was approximately 4.5 wt%.: How did you quantify the final amount of SF in an aqueous solution? Any purity result?

Table 1. Reaction points in silk fibroin: It is not clear, the final concentration of SF in reaction mixture. If molar ratio, then how did the authors mix 20 molar from the final concentration of 4.5 wt% (Line 101) in the reaction mixture?

Figure 1.: mention the Tyr residues in SF, also mention the intermediate molecule release from the reaction after crosslink. What about the possibilities of cross-reaction of modified heparin with side-chain (CH2-CH-NH(CO)-) of SF? 

heavy water as a lock solvent: What is lock solvent?

2,4,6-trichloro-1,3,5-triazine (CY) : Mention the other name Cyanuric chloride.

a cellulose tube, and dialyzed against purified water: Specification of dialysis membrane.

e purchased and cultured with Clonetics™: City and country name

™ cell systems EBM™ 2 (Lonza), which contains endothelial...and FBM-BulletKit™ which contains: Split this sentence into two or three to get a clear point.

2.4. Secondary structure analysis by solid-state 13C CP/MAS NMR: WHy specifically used SF and HSF5, why not other samples (HSF 2-4)

2.5. Heparin coverage on the film surface evaluation: How did you make film from the composite? Explain the method and sample details.

Cast film samples were prepared from each HSF aqueous solution: How? Explain the method

Its not clear sample types of samples used in all experiments, some with sponges and some experiment with casting films, and some with an aqueous liquid. 

Solid-state 13C CP/MAS NMR measurements were carried out on the untreated film: Why only untreated film? What is an untreated film, and how it was made?

for the insolubilized film subjected: What is insolubilized film?

high humidity treatment under 100% relative humidity: Explain the method clearly, since this method was used in the following section 2.5 as well.

high humidity insolubilization treatment:Explain the method briefly. 

The other elements on each insolubilized film's surfaces: What do you mean by other elements here?

2.5. Heparin coverage on the film surface evaluation: Explain the method of how the Heparin was covered on the film surface?

a thick cast film made following the procedure was used to set the instrument. Each HSF sponges: Make clear sample type. is it cast film or sponge?

HSF sponges were dissolved in formic acid: concentrated formic acid?

The cast films were incubated at 37°C and 100 % relative humidity: Explain how you achieved 100% relative humidity.

pasted on a glass substrate: How using any glue?

insolubilized vacuum dried samples: What exactly the sample is? what is insolubilized? 

 dry and wet films were calculated: How did you measure the initial dry weight and wet weight of films?

The percentage water content: Double check the formula, seems like [(Wet Wt-Dry Wt)/Dry Wt]x 100

 0.5 ml VEGF, 0.5 ml hEGF, 0.5 ml R3-IGF-1: Expand them

Cellular responses to each HSF sample: List out the sample, what was the final state of these samples? powder or liquid? 

It was prepared into a 24-well plate: What was prepared here? 

 to a thickness of 0.25 mg/cm2:How did you control the thickness? how it was measured? Explain these method details.

film swollen with PBS: any sterilization before cell seeding?

1.0 × 103 cells/mL, whereas NHDF was seeded at 0.5 × 103: Why different cell densities?

with a fresh medium every few days: It's better to say precisely. 

, the cultures were maintained for up to 7 days in NHDF: Not clear. How did you maintain the cell cultures in NHDF cells?

 7 days in NHDF: What about HUVEC?

Figure 2:Provide axis in all images

Figure 2 The spectra comparing (b) the Tyr residue: Is it NMR or UV spectra?

Figure 3:Provide axis

Figure 4. SEM-EDS images: Very poor resolution images

 (e) shows the water contact angle: e or f? better to provide some images of the water contact angle experiment 

Figure 5.The swelling ratios of SF: Explain why HSF4 had higher water content..

Use consistent name throughout MS either Neo-NHDF or  NHDF

Figure 7. The proliferation: Why 3 days for  HUVEC and 7 days for Neo-NHDF

The proliferation of (a) HUVEC and (b) Neo-NHDF:Explain the reason why  HSF3, HSF4, and HSF5 stimulated cell proliferation on day 3 for HUVEC and on day 7 for Neo-NHDF (not day 3 and 5)?

A good wound dressing provides cells: The present study results did not support/evidence the wound dressing effect.

by the active capture and donation of GFs: What types of GFs released in this study to support the two cells' proliferation by HSF samples.

SF promotes the longterm proliferation of cells: The study results did not evidence this statement, because the cells were cultured with HSF only for a short time 3 and 7 days. 

SF-based materials as wound dressings to promote tissue regeneration.: This study did not deal with tissue regeneration, it may increase cell proliferation, but does not mean can support tissue regeneration.

Author Response

Dear Reviewer:

We appreciate your kind and insightful comments on our manuscript. We have carefully reviewed the comments and have revised the manuscript accordingly and corrected the grammatical issues pointed out in the manuscript and accompanying figures with our best to make it better.
Our responses are given in a point-by-point manner below. Changes to the manuscript are shown in red. The comments are numbered consecutively.
We hope the revised version is now suitable for publication and look forward to hearing from you in due course.

  1. heparin exposed on the film surface correlated with the modification reaction, and water wettability: Provide the data here.

Response:
The expression was modified to be more specifically.
"Increased amounts of heparin for modification raised heparin content and water wettability on film surfaces."

  1. In addition, cell proliferation is promoted; Not clear which cells

Response:
We added notation for vascular endothelial cells and fibroblasts.
"proliferations of endothelial cell and fibroblast were enhanced when the surface with sufficient amounts of heparin"

  1. cell proliferation is promoted to the same extent: same extent compared to ?

Response:
We changed expression to be more clear.
”proliferation was enhanced in both cells when the surface with sufficient amounts of heparin”

  1. with a sufficient amount of heparin, demonstrating its usefulness in assisting wound healing: Simply proliferation data did not claim the wound healing behaviour, should be evidenced by some other experiments like scratch assay and other experiments...

Response:
We changed expression to be more correct.
”been expected its potential in assisting wound healing.”

  1. In applying SF in biomaterials: What do you mean biomaterials here? SF is a biomaterial

Response:
In this explanation, biomaterial was intended to mean "tissue engineering material," so it was corrected to avoid confusion.
" tissue enigineering material"

  1. characteristic crystal structures from a multifaceted evaluation: Present the results in abstract.

Response:
We changed expression to be clearer and more specific.
”This research emphasizes the importance of material design focusing on the crystal structure inherent to SF in the development of functionalized SF materials.”

  1. research has been reported by: Give reference

Response:
We added references [6] and [7].
[6] Idrus, R.B.H.; Rameli, M.A.B.P.; Cheong, L.K.; Xian, L.J.; Hui, C.K.; Latiff, M.B.A.; Saim, A. Bin Allogeneic Bilayered Tissue-Engineered Skin Promotes Full-Thickness Wound Healing in Ovine Model. Biomed. Res. 2014, 25, 192–198.
[7] Millán-Rivero, J.E.; Martínez, C.M.; Romecín, P.A.; Aznar-Cervantes, S.D.; Carpes-Ruiz, M.; Cenis, J.L.; Moraleda, J.M.; Atucha, N.M.; García-Bernal, D. Silk Fibroin Scaffolds Seeded with Wharton’s Jelly Mesenchymal Stem Cells Enhance Re-Epithelialization and Reduce Formation of Scar Tissue after Cutaneous Wound Healing. Stem Cell Res. Ther. 2019, 10, 1–14, doi:10.1186/s13287-019-1229-6.

  1. was selected as the base material: In this study or earlier reference?

Response:
In this study, SF was selected as a base material to modify a functional molecule (heparin). The wording was changed.
"silk fibroin (SF), which has mild and controllable, biodegradable[14], exhibits low inflammation[15], and promotes skin regeneration[16–18], was selected as a base material."

  1. Several studies have been published on combining heparin and SF[21]: Provide a few more references here.

Response:
We added references.
“Several studies have been published on combining heparin and SF[23–25]. Cell proliferation has been shown to improve on non-woven fabrics by electrospinning of heparin and SF solutions[26], and a sponge with heparin physically immobilized on the SF has been reported[25].”
[24] Seib, F.P.; Herklotz, M.; Burke, K.A.; Maitz, M.F.; Werner, C.; Kaplan, D.L. Multifunctional Silk-Heparin Biomaterials for Vascular Tissue Engineering Applications. Biomaterials 2014, 35, 1–20, doi:10.1016/j.biomaterials.2013.09.053.Multifunctional.
[25] Çakır, C.O.; Ozturk, M.T.; Tuzlakoglu, K. Design of Antibacterial Bilayered Silk Fibroin-Based Scaffolds for Healing of Severe Skin Damages. Mater. Technol. 2018, 33, 651–658, doi:10.1080/10667857.2018.1492209.

  1. concentration of the purified SF aqueous solution was approximately 4.5 wt%.: How did you quantify the final amount of SF in an aqueous solution? Any purity result?

Response:
It was calculated by measuring the weights of substances obtained by completely drying SF solution after dialysis. This procedure was added there. Also, we didn't measure purity in this study.
"The final concentration of the purified SF aqueous solution was approximately 4.5 wt% by measuring the weight of substrate obtained by drying the SF solution after the end of dialysis.”

  1. Table 1. Reaction points in silk fibroin: It is not clear, the final concentration of SF in reaction mixture. If molar ratio, then how did the authors mix 20 molar from the final concentration of 4.5 wt% (Line 101) in the reaction mixture?

Response:
They were calculated based on references [29, 30]. SF used an average molecular weight of 370,000 for the Heavy chain and the number of Tyr and Lys residues in the amino acid sequence. Heparin used an average molecular weight of 15,000. We added to the text. "It was calculated from the amount of Tyr and lysine (Lys) residues in the SF heavy chain and the average molecular weight of heparin of 15,000."

  1. Figure 1.: mention the Tyr residues in SF, also mention the intermediate molecule release from the reaction after crosslink. What about the possibilities of cross-reaction of modified heparin with side-chain (CH2-CH-NH(CO)-) of SF?

Response:
The activated chlorine in CY reacts with hydroxyl and amino groups in proteins. In the acidic region below the isoelectric point of silk (approx. pH 4.0), side reactions with N-terminal amines in addition to Lys residues may occur. However, to avoid this, we set the pH in the alkaline region above the isoelectric point of the Tyr residue (approx. pI 10). Also, even thou we calculated; these side reactions did not affect the significant figures for the molar ratios shown in Table 1.

  1. heavy water as a lock solvent: What is lock solvent?

Response:
Solution NMR measurements require adding a deuterated solvent for frequency locking. In this experiment, a small amount of D2O was added to each aqueous solution sample. To avoid confusion, the description in section 2.3 has been changed.

"with D2O."

  1. 2,4,6-trichloro-1,3,5-triazine (CY) : Mention the other name Cyanuric chloride.

Response:
The notation has been added before the abbreviation. We appreciate your kind remarks about these inadequate notations.
"(Cyanuric Chloride, CY, Kanto Chemical CO., INC. Tokyo, Japan)"

  1. a cellulose tube, and dialyzed against purified water: Specification of dialysis membrane.

Response:
Product sizes and suppliers were added, and additional information was added for other materials.
"size 21.4, Nihon Medical Science, Osaka, Japan"

  1. purchased and cultured with Clonetics™: City and country name
  2. ™ cell systems EBM™ 2 (Lonza), which contains endothelial...and FBM-BulletKit™ which contains: Split this sentence into two or three to get a clear point.

Response:
In response to points 16 and 17, we have organized Lonza's material information used for cell culture and revised the text.
"HUVEC and NHDF lines (Lonza, MD, USA) were purchased. HUVECs were grown in HUVEC-XL™ cell systems EBM™ 2 (Lonza) that contained 2 %(v/v) fetal bovine se-rum (FBS), 0.5 ml vascular endothelial growth factor (VEGF), 0.5 ml human epidermal growth factor (hEGF), 0.5 ml insulin-like growth factor-1 with the substitution of argi-nine for glutamic acid at position-3 (R3-IGF-1), 0.5 ml ascorbic acid, 0.2 ml hydrocortisone, 2 ml human fibroblast growth factor-base (hFGF-β), 0.5 ml gentamicin/amphotericin-B (GA), and 0.5 ml heparin. NHDFs were cultured with FBM-BulletKit™ (Lonza) sup-plemented with 2 %(v/v) FBS, 0.5 ml hFGF-β, 0.5 ml insulin, and 0.5 ml GA."

  1. 4. Secondary structure analysis by solid-state 13C CP/MAS NMR: WHy specifically used SF and HSF5, why not other samples (HSF 2-4)
  2. Solid-state 13C CP/MAS NMR measurements were carried out on the untreated film: Why only untreated film? What is an untreated film, and how it was made?
  3. for the insolubilized film subjected: What is insolubilized film?
  4. insolubilized vacuum dried samples: What exactly the sample is? what is insolubilized?

Response:

In response to points 18,22,23,32, the crystalline structure of SF was induced by insolubilization treatment and this structural transition was observable by solid-state NMR measurements. This was done to inspect the possibility that the modification of heparin would inhibit the formation of the crystal structure of SF. Therefore, HSF5, which was found to have the highest amount of modification in aqueous HSF solutions, was selected for comparison. As shown in the caption of Figure 3, three different samples were used: SF-T (after insolubilization), HSF5-N (before insolubilization), and HSF5-T (after insolubilization). Untreated (before insolubilization) refers to film obtained by casting and air-drying, while insolubilized refers to film in which structural transition was induced by the high-humidity treatment method. The description has been added to the text.
"13C CP/MAS NMR spectra of film samples of SF-T, HSF5-N, and HSF5-T films (T: after insolubilized, and N: before insolubilized). Peak intensities were normalized by SF glycine Cα."

  1. 5. Heparin coverage on the film surface evaluation: How did you make film from the composite? Explain the method and sample details.
  2. Cast film samples were prepared from each HSF aqueous solution: How? Explain the method
  3. Its not clear sample types of samples used in all experiments, some with sponges and some experiment with casting films, and some with an aqueous liquid.
  4. Solid-state 13C CP/MAS NMR measurements were carried out on the untreated film: Why only untreated film? What is an untreated film, and how it was made?
  5. a thick cast film made following the procedure was used to set the instrument. Each HSF sponges: Make clear sample type. is it cast film or sponge?

Response:
This is in response to points 19-22,28. Solution samples were used only for solution NMR, cast films were used for the other film structure and characterization. The method of dissolving sponges obtained by freeze-drying aqueous solutions in organic solvents at high concentrations was chosen as a means of producing cast films with thicker thickness because the concentration of the obtained aqueous HSF solutions was very low, less than 1 wt%. In the cell studies, very thin cast films were prepared on each well of cell culture plates to account for the change in stiffness of the HSF films due to water content. To avoid confusion, the terminology used in the film preparation method descriptions in each section has been reorganized and corrected.

"2.4: Solid-state NMR measurements of films, before (as-cast) and after insolubilization with high humidity, were performed to examine the changes in the secondary structure of SF in HSF[29]. In these experiments, SF and HSF5 with the highest heparin modification were selected. Thick cast films for film characterization were obtained by the following procedure: sponges by lyophilized of SF and HSF aqueous solutions were dissolved in 1,1,1,3,3,3-hexafluoro-2-propanol (HFIP) and cast on a Teflon dish with a diameter of 50 mm and air-dried, then incubated in a container with about 100% relative humidity at 37°C for 24 h."

2.5. Evaluation of film surface properties “HSF aqueous solutions were cast on glass plates and air dried at room temperature, then incubated in high humidity at 37 ℃ for insolubilization for 24 h.” “For water contact angle measurements, thick cast films were made following a procedure to set the instrument. Each HSF sponge was dissolved in 98% formic acid (FA, FUJIFILM) at 6.0 wt%, then 3.0 mL of each solution was cast on polystyrene dishes and air-dried at room temperature. The cast films were incubated at 37°C and 100 % relative humidity for 24 h for insolubilization, hollowed out and pasted on a glass substrate with double-sided tape.”

  1. high humidity treatment under 100% relative humidity: Explain the method clearly, since this method was used in the following section 2.5 as well.
  2. The cast films were incubated at 37°C and 100 % relative humidity: Explain how you achieved 100% relative humidity.

Response:
This is in response to points 24 and 30. We have corrected this representational issue. The general method of incubating a sealed container with a small amount of ultrapure water was used. Changes in relative humidity over time were measured and confirmed that high humidity was maintained.

"then incubated in a container with about 100% relative humidity at 37°C for 24 h."

  1. high humidity insolubilization treatment: Explain the method briefly.

Response:
Notation was also added to 2.5.
"HSF aqueous solutions were cast on glass plates and air dried at room temperature, then incubated in high humidity at 37 ℃ for insolubilization. Elements on insolubilized film surfaces were evaluated."

  1. The other elements on each insolubilized film's surfaces: What do you mean by other elements here?

Response:
There was a problem with the English description. In this experiment, elemental analysis by SEM-EDS was performed for S and Na.
"Elements on each insolubilized film's surfaces were evaluated by SEM-EDS analysis with an energy dispersive X-ray analyzer Quantax 70, using scanning electron microscopy (SEM) Miniscope® TM3000 (Hitachi High-Technologies, Corp.)."

  1. 5. Heparin coverage on the film surface evaluation: Explain the method of how the Heparin was covered on the film surface?

Response:
Since we have not measured the structure of the film surface by AFM measurements in this experiment, we have not been able to determine the structure of heparin exposed on the film surface. We observe an increase in the abundance of S elements derived from heparin molecules and an increase in hydrophilicity, indirectly confirming the presence of heparin on the film surface. We have changed the title of section 2.5.
"2.5. Evaluation of film surface properties"

  1. HSF sponges were dissolved in formic acid: concentrated formic acid?

Response:
Concentrated formic acid (98.0+% grade) is commonly used as a solvent for SF. We added the concentration. "dissolved in 98% formic acid (FUJIFILM)"

  1. pasted on a glass substrate: How using any glue?

Response:
Added notation "double-sided tape was used.
"pasted on a glass substrate with double-sided tape."

  1. insolubilized vacuum dried samples: What exactly the sample is? what is insolubilized?
  2. dry and wet films were calculated: How did you measure the initial dry weight and wet weight of films?

Response:
In response to points 32,33, we corrected the description of samples and measurements in section 2.6.
"The water content of films (SF and HSFs) was measured. Each SF and HSF aqueous solution was cast on glass plates and air dried at room temperature. After high humidity insolubilization, films were vacuum dried 3 h and measured as dry weight. Then they were immersed in purified water for 3 h and measured again as wet weight. The water content was calculated using equation 1 below. "

  1. The percentage water content: Double check the formula, seems like [(Wet Wt-Dry Wt)/Dry Wt]x 100

Response:
We corrected it as noted.
"[(wet weight – dry weight)/dry weight] × 100"

  1. 5 ml VEGF, 0.5 ml hEGF, 0.5 ml R3-IGF-1: Expand them

Response:
We corrected the notation.
"0.5 ml vascular endothelial growth factor (VEGF), 0.5 ml human epidermal growth factor (hEGF), 0.5 ml insulin-like growth factor-1 with the substitution of arginine for glutamic acid at position-3 (R3-IGF-1), 0.5 ml ascorbic acid, 0.2 ml hydrocortisone, 2 ml human fibroblast growth factor-base (hFGF-β), 0.5 ml gentamicin/amphotericin-B (GA)"

  1. Cellular responses to each HSF sample: List out the sample, what was the final state of these samples? powder or liquid?
  2. It was prepared into a 24-well plate: What was prepared here?

Response:
In response to 36,37, we have corrected the ambiguity in the wording. Here, coated thin films were prepared on 24-well plates for cell culture, and then cells were seeded.
"Each aqueous cast film of SF and HSFs were prepared into 24-well plates at 0.25 mg/cm2,"

  1. to a thickness of 0.25 mg/cm2:How did you control the thickness? how it was measured? Explain these method details.

Response:
The term "thickness" has been corrected to avoid confusion.
"into 24-well plates at 0.25 mg/cm2"

  1. film swollen with PBS: any sterilization before cell seeding?

Response:
Sterilization was performed with 70% ethanol.
"immersed in 70 % (v/v) ethanol solution for 30 min for insolubilization and sterilization"

  1. 0 × 103 cells/mL, whereas NHDF was seeded at 0.5 × 103: Why different cell densities?

Response:
Each cell type has a different level of metabolic activity per cell. Seeding densities that were sufficiently detectable and did not achieve excessive confluency early in the culture period were preliminarily investigated.

  1. with a fresh medium every few days: It's better to say precisely.

Response:
It was corrected to "every 2~3 days" to be more specific.
"and each medium was replaced with a fresh one every two or three days."

  1. , the cultures were maintained for up to 7 days in NHDF: Not clear. How did you maintain the cell cultures in NHDF cells?
  2. 7 days in NHDF: What about HUVEC?

Response:
In response to points 42 and 43, we corrected notations.
"NHDF was incubated for up to seven days for investigate long-term growth behaviors, and HUVEC was incubated for three days."

  1. Figure 2:Provide axis in all images
  2. Figure 2 The spectra comparing (b) the Tyr residue: Is it NMR or UV spectra?
  3. Figure 3:Provide axis

Response:
This is in response to points 44-46. These NMR spectra were normalized by peaks derived from SF and arranged. Therefore, we added the normalization conditions instead of the axes.
"Figure 2. 1H NMR spectra of SF and HSF aqueous solutions. Every concentration was 0.5 wt% and peak intensities were normalized by silk fibroin (SF) alanine (Ala) Hβ. (a) SF, heparin, and heparin-modified SF samples (HSF2, HSF3, HSF4, and HSF5) from top to bottom. (b) Spectra around SF tyrosine (Tyr) residues with a structural formula of a Tyr residue. (c) Spectra around a CH group region which contains heparin CH, SF serine (Ser) and Tyr residues demonstrated with different colors from bottom to top."
"Figure 3. 13C CP/MAS NMR spectra of silk fibroin-T, HSF5-N, and HSF5-T films (T: in-solubilized, and N: before insolubilized samples). Peak intensities were normalized by SF glycine Cα."

  1. Figure 4. SEM-EDS images: Very poor resolution images

Response:
We corrected the image resolution.

  1. (e) shows the water contact angle: e or f? better to provide some images of the water contact angle experiment

Response:
Figure description was revised to (f) and photos of water droplets were added (g,h).
"SEM-EDS images of the distributions of S and Na on the surface of SF (a), HSF2 (b), HSF3 (c), HSF4 (d), and HSF5 (e) films. (f) shows a water contact angle of SF and HSF samples ( g: SF, h: HSF5)."

  1. Figure 5.The swelling ratios of SF: Explain why HSF4 had higher water content..

Response:
Multiple runs showed no statistically significant differences between HSF4 and HSF5. Possible considerations at this time include the steric structure of heparin between SF crystal structures and the interaction between HSF and water molecules, as discussed in the Discussion section.

  1. Use consistent name throughout MS either Neo-NHDF or NHDF

Response:
We unified the wording NHDF.

  1. Figure 7. The proliferation: Why 3 days for HUVEC and 7 days for Neo-NHDF
  2. The proliferation of (a) HUVEC and (b) Neo-NHDF: Explain the reason why HSF3, HSF4, and HSF5 stimulated cell proliferation on day 3 for HUVEC and on day 7 for Neo-NHDF (not day 3 and 5)?

Response:
This is in response to points 51,52. From these cell experiments, we consider that the time at which cells bind GFs and begin to proliferate and appear in the metabolic activity assay was dependent on the metabolic activity characteristics of the cell type. In the future, we plan to evaluate gene and protein expression in order to analyze the effects on cell proliferation through GF receptors in detail.

  1. A good wound dressing provides cells: The present study results did not support/evidence the wound dressing effect.

Response:
We corrected wording.
”Good wound dressings have to be possible to provide cells "

  1. by the active capture and donation of GFs: What types of GFs released in this study to support the two cells' proliferation by HSF samples.

Response:
Cells are affected by multiple GFs because both media of NHDF and HUVEC contain FBS. However, these effects are considered to be particularly significant because of the addition of GFs as supplements added to each medium. Therefore, we added these GFs to the explanation.
"Furthermore, it is thought that cell proliferation was strongly promoted by the active capture and donation of GFs, such as FGF and VEGF."

  1. SF promotes the longterm proliferation of cells: The study results did not evidence this statement, because the cells were cultured with HSF only for a short time 3 and 7 days.

Response:
The notation was corrected because of the short cultivating period in this study.
"heparin-modified SF promoted proliferation of cells"

  1. SF-based materials as wound dressings to promote tissue regeneration.: This study did not deal with tissue regeneration, it may increase cell proliferation, but does not mean can support tissue regeneration.

Response:
We fixed a leap in wording.
"It provides the basic knowledge for enabling design strategies that take advantage of characteristics of SF-based materials for wound dressings."

Reviewer 2 Report

 More relevant references related with silk interaction with heparin e.g. such as Seib FP, Herklotz M, Burke KA, Maitz MF, Werner C, Kaplan DL. Multifunctional silk-heparin biomaterials for vascular tissue engineering applications. Biomaterials. 2014 Jan;35(1):83-91. doi:10.1016/j.biomaterials.2013.09.053.

could be discussed in Introduction with special additional attention to the role of heparin

It would be beneficial to more specifically and clearly formulate the research objective at the end of Introduction and estimate novelty and perspectives of application of research results in Conclusions. 

 In Section 2. Materials and Methods characteristics of heparin must be presented in more detail, e.g. molecular weight of heparin.

In the Journal Arteriosclerosis, Thrombosis, and Vascular Biology. 2003;23:2110–2115 Thorama and coworkers published article Heparin Inhibition of endothelial cell proliferation and organization is dependent on molecular weight.  The inhibition of endothelial cell proliferation is in contradiction with your statement in line 315 that Heparin induces vascular endothelial cell and fibroblast proliferation, but no comments are provided. It is known that cell proliferation strongly depends on substrate stiffness,  but such dependence was not discussed in the article. The provided in Discussion section possible mechanisms have not been clearly and duly justified.

The role of water briefly mentioned in lines 380-382 needs to be discussed in much more clearly and in more detail.

Author Response

Dear Reviewer :
We appreciate your kind and insightful comments on our manuscript. We have carefully reviewed the comments and have revised the manuscript accordingly and corrected the grammatical issues pointed out in the manuscript and accompanying figures with our best to make it better.
Our responses are given in a point-by-point manner below. Changes to the manuscript are shown in red. The comments are numbered consecutively.
We hope the revised version is now suitable for publication and look forward to hearing from you in due course.

  1. More relevant references related with silk interaction with heparin e.g. such as Seib FP, Herklotz M, Burke KA, Maitz MF, Werner C, Kaplan DL. Multifunctional silk-heparin biomaterials for vascular tissue engineering applications. Biomaterials. 2014 Jan;35(1):83-91. doi:10.1016/j.biomaterials.2013.09.053.

Response:
References including this have been added.

[24] Seib, F.P.; Herklotz, M.; Burke, K.A.; Maitz, M.F.; Werner, C.; Kaplan, D.L. Multifunctional Silk-Heparin Biomaterials for Vascular Tissue Engineering Applications. Biomaterials 2014, 35, 1–20, doi:10.1016/j.biomaterials.2013.09.053.Multifunctional.
[25] Çakır, C.O.; Ozturk, M.T.; Tuzlakoglu, K. Design of Antibacterial Bilayered Silk Fibroin-Based Scaffolds for Healing of Severe Skin Damages. Mater. Technol. 2018, 33, 651–658, doi:10.1080/10667857.2018.1492209.

  1. could be discussed in Introduction with special additional attention to the role of heparin

Response:
Heparin was used to strongly improve the cellular response of SF by capturing and supplementing GF. We revised the notation.

"It binds to various major GFs: vascular endothelial growth factor (VEGF)[9], fibroblast growth factor (FGF)[10,11], epidermal growth factor (EGF)[12], and hepatocyte growth factor (HGF)[13]. Thus, heparin was expected to act as a functional molecule that specifically captured secreted GFs into the material during the healing process."

  1. It would be beneficial to more specifically and clearly formulate the research objective at the end of Introduction and estimate novelty and perspectives of application of research results in Conclusions.

Response:
As pointed out, we have revised our views in the Introduction and Conclusion.

"To consider this hypothesis, we prepared materials with varying amounts of heparin modification and evaluated the effect of heparin modification on the structure of SF and its stability in water by comparing these materials. In addition, the interaction with epithelial cells and fibroblasts was evaluated for application to skin wound dressings."

"In this study, we conducted a wide-range evaluation focusing on the relationship be-tween the material and water for HSF films that are stable in water and whose modification amount is gradually changed. It's possible to develop functional materials whose physical properties and cellular responsiveness can be more precisely controlled by focusing on the crystal structure characteristic of SF to modify. This provides the basic knowledge for enabling design toward induction material-driven wound healing."

  1. In Section 2. Materials and Methods characteristics of heparin must be presented in more detail, e.g. molecular weight of heparin.

Response:
As mentioned above, we have changed the whole wording of the film material preparation method and added the heparin molecular weight used in the heparin modification reaction.

  1. In the Journal Arteriosclerosis, Thrombosis, and Vascular Biology. 2003;23:2110–2115 Thorama and coworkers published article Heparin Inhibition of endothelial cell proliferation and organization is dependent on molecular weight. The inhibition of endothelial cell proliferation is in contradiction with your statement in line 315 that Heparin induces vascular endothelial cell and fibroblast proliferation, but no comments are provided. It is known that cell proliferation strongly depends on substrate stiffness, but such dependence was not discussed in the article. The provided in Discussion section possible mechanisms have not been clearly and duly justified.

Response:
Thank you for sharing the interesting article and comments. Assuming that the structure and molecular weight of unfractionated heparin were comparable between this paper and our experiments, we think that the differences in experimental systems had different effects on cell proliferation. Even though both studies used unfractionated heparin, whereas Khorana et al. added heparin to the medium and we used it as a scaffold for cell adhesion. This difference in experimental systems is important because it oppositely changes the direction of the interaction between cells and heparin. In the discussion by citation 24, it was suggested that when the molecular weight of heparin was large, steric hindrance prevented dimerization with the cell surface receptor. In other words, heparin in the medium might bind in a lower ratio to the cells adhering to the culture dish and inhibit the process of forming a complex. On the other hand, we have cells adhering to the surface of scaffolds (heparin-modified silk film) on which heparin was exposed. Therefore, the effect of steric hindrance due to the large molecular weight of heparin was decreased, and the cell proliferation effect by the complex formation of "heparin - growth factors - cell surface receptors" was considered to be stronger than this negative effect. In addition, we also concerned about the effect of film stiffness on cell behavior as you pointed, cell culture films were prepared as a thin coating (0.25 mg/cm2) on 24-well plates. Description added in 2.9, and Discussion section.

  1. The role of water briefly mentioned in lines 380-382 needs to be discussed in much more clearly and in more detail.

Response:
We revised the explanation. "These increases in hydrophilicity on the surface and inside the material were expected to work not only to improve the adhesion of migrating cells to the material but also to in-crease the adsorption capacity of GFs through exudate absorption and promoted the proliferation of vascular endothelial cells and fibroblasts."

Round 2

Reviewer 1 Report

The revised version is satisfactory and pleased to accept it.

Reviewer 2 Report

I agree with the authors comments and corrections